# Identification of Critical Genes and Pathways for Influenza A Virus Infections via Bioinformatics Analysis

**DOI:** 10.3390/v14081625

**Published:** 2022-07-26

**Authors:** Gao Chen, Haoyue Li, Mingzhao Hao, Xiaolei Li, Yizhi Dong, Yue Zhang, Xiping Liu, Cheng Lu, Jing Zhao

**Affiliations:** 1School of Life Science, Hubei University, Wuhan 430062, China; gaochen_npwr@hotmail.com; 2Institute of Basic Research in Clinical Medicine, China Academy of Chinese Medical Sciences (CACMS), Beijing 100700, China; hao911815@126.com (H.L.); dr_lixiaolei@163.com (X.L.); dongyizhi1997@163.com (Y.D.); zhangyue950309@163.com (Y.Z.); a1132692895@163.com (X.L.); lv_cheng0816@163.com (C.L.); 3Institute of History of Medicine and Medical Literature, China Academy of Chinese Medical Sciences, Beijing 100700, China; mingzhao_h@163.com

**Keywords:** influenza A virus, hub gene, bioinformatics, weighted gene co-expression network analysis, protein interaction

## Abstract

Influenza A virus (IAV) requires the host cellular machinery for many aspects of its life cycle. Knowledge of these host cell requirements not only reveals molecular pathways exploited by the virus or triggered by the immune system but also provides further targets for antiviral drug development. To uncover critical pathways and potential targets of influenza infection, we assembled a large amount of data from 8 RNA sequencing studies of IAV infection for integrative network analysis. Weighted gene co-expression network analysis (WGCNA) was performed to investigate modules and genes correlated with the time course of infection and/or multiplicity of infection (MOI). Gene Ontology (GO) and Kyoto Encyclopedia of Genes and Genomes (KEGG) enrichment analyses were performed to explore the biological functions and pathways of the genes in 5 significant modules. Top hub genes were identified using the cytoHubba plugin in the protein interaction network. The correlation between expression levels of 7 top hub genes and time course or MOI was displayed and validated, including BCL2L13, PLSCR1, ARID5A, LMO2, NDRG4, HAP1, and CARD10. Dysregulated expression of these genes potently impacted the development of IAV infection through modulating IAV-related biological processes and pathways. This study provides further insights into the underlying molecular mechanisms and potential targets in IAV infection.

## 1. Introduction

The influenza A virus (IAV), a member of the Orthomyxoviridae family, is the causal agent of an acute respiratory tract infection suffered annually by 5–20% of the human population [1]. IAV can cause high mortality in humans, with 250,000–500,000 deaths per year worldwide [2]. Current treatments are focused on vaccines and drugs that target viral proteins. However, both of these approaches have limitations as vaccines require annual development to match the antigenic strains circulating, while viral proteins have an impressive capacity to evolve resistance against anti-viral agents [3]. With the expression of 14 functional proteins for viral replication and virulence, the repertoire of gene products on the pathogen side is limited. The viral life cycle and the replication of the IAV are dependent on hijacking host-cell biological processes to facilitate entry, replication, assembly, and budding. The recognition that a suite of mammalian host proteins is required for IAV infection and replication presents additional targeting strategies that may be less prone to deflections by the quickly mutating viral genome.

Influenza infection activates a number of host pathways, including the innate and adaptive immune responses, the induction of cytokines, and the activation of apoptosis [4]. The detection of viral particles (in particular nuclear acids) by toll-like receptors (TLR7) of the MyD88, NF-kB pathway [5], as well as cytosolic proteins such as RIG-I (DDX58) of the MDAF/MAVS pathway and their trigger of interferon expression via the activation of transcription factors including IRF3 and IRF7 [6,7], have been well studied. Unfortunately, much less is known about downstream host factors and pathways.

In recent years, the amount of biological data for research has dramatically increased with the development of transcriptome analysis, providing new viewpoints for exploring the pathogenesis and novel targets for clinical treatment [8,9,10,11]. Previous studies primarily tested individual genes in diseased conditions. However, genes with similar expression patterns are likely to be tightly co-regulated in vivo with closely related functions, expressed in the same signaling pathways or processes [12]. Identifying these genes will provide a further understanding of biological pathways in the host’s response to IAV infection.

In the present study, multiple bioinformatics methods were used to search for top hub genes in the biological pathways that were significantly correlated with the time course and multiplicity of infection (MOI) for IAV infection. These correlations have also been validated in a dataset from another independent experiment. This work will provide further insights into the underlying molecular mechanisms of the host response to IAV infection and potential molecular targets for developing novel interventional strategies.

## 2. Materials and Methods

### 2.1. Dataset and Preprocessing

In this study, eight RNA sequencing files for IAV infection containing GSE165340, GSE97672, GSE104168, GSE156152, GSE163959, GSE193164, GSE186908, and GSE89008 were obtained from the Gene Expression Omnibus (GEO) repository (https://www.ncbi.nlm.nih.gov/geo/, accessed on 5 May 2022). The main features of these 8 datasets were shown in Table 1 (Appendix A). These profiles consisted of gene expression matrices and experiment design. Subsequent analyses were conducted on these datasets.

All of the raw data of GSE165340, GSE97672, GSE104168, GSE156152, GSE163959, GSE193164, and GSE186908 were merged and normalized into the training group through a unified ENCODE RNA-seq processing pipeline, followed by removing the batch effect using the Limma package of R [13]. The boxplot was used to estimate whether the batch effect was removed (Appendix A). The validation group was the dataset from GSE89008.

### 2.2. Identification of Significant Modules Using the Weighted Gene Co-Expression Network Analysis

The WGCNA package [14] was used to determine key genes significantly associated with IAV infection in the training and validation groups. The best soft threshold power was set to identify the module–trait relationship, module membership (MM), and gene significance (GS). In brief, a weighted adjacency matrix was first constructed based on the selected soft threshold power. Subsequently, the connectivity per gene was deduced through calculating connection strengths with other genes. After validating the module structure preservation using the module preservation R function, the gene expression profile of each module was summarized by the module eigengene on whom the designated traits were regressed in the Limma R package [15]. The IAV-related module was selected with the *p*-value < 0.05.

### 2.3. Function Enrichment Analysis

An online tool (https://david.ncifcrf.gov/, accessed on 5 May 2022) was employed to perform a Function Enrichment Analysis of the genes in the significant modules [16]. The Gene Ontology (GO) [17] and Kyoto Encyclopedia of Genes and Genomes (KEGG) [18] analysis was conducted for the exploration of the involved biological functions and pathways of the above genes. The items ranked within the top 10 according to *p*-value were selected and visualization was conducted using the barplot function in R software.

### 2.4. Protein–Protein Interaction Network Analysis

The Search Tool for the Retrieval of Interacting Genes (The Human Reference Interactome and Literature Benchmark, HuRI, http://interactome-atlas.org, accessed on 5 May 2022) [19] was used to identify interactions between the products of the genes in the modules. The protein–protein interaction (PPI) network was constructed using HuRI by adopting the default threshold and visualized using Cytoscape 3.8.2. The connection degree (number) of each node was calculated using the cytoHubba plugin [20] within Cytoscape.

### 2.5. The Correlation between Expression Levels of Hub Genes and Traits

The ggcorrpolt and the ggthemes of the R package were used to determine the correlation between expression levels of hub genes and time course or MOI in the training and validation groups, respectively. *p*-value < 0.05 was considered statistically significant.

## 3. Results

### 3.1. Identification of IAV-Associated Modules

A total of 278 samples and 12,103 genes retrieved from the training group were used for the co-expression network analysis (Figure 1). An eigengene correlation coefficient square and a soft threshold power of 4 (Appendix A) were set to identify gene modules. In order to resolve computational challenges caused by constructing and analyzing networks with such large numbers of nodes, function blockwiseModules in the WGCNA package was used to split into two blocks for hierarchical clustering. The hierarchical clustering trees were constructed following a dynamic hybrid cut (Figure 1A,B). Thirty-four were identified when the DissThres was set as 0.25 after merging dynamic modules, as shown in the clustering dendrograms (Figure 1C). The eigengene dendrogram and heatmap were used to quantify module similarity by eigengene correlation (Figure 1C,D).

The distribution of module eigengene change in each experiment design of the training group showed that the module eigengene change was an upward trend along with the value of time or MOI (Figure 2), suggesting that there was a correlation between time and/or MOI and modules. Among the 7 studies in the training group, all of the IAV strains and most of the cell types were different (Table 1). A noticeable feature was not displayed between time and/or MOI and modules in Figure 2. Further analysis confirmed that 5 gene modules were significantly correlated with the time course and/or MOI (Figure 3, Appendix A). The salmon module was positively correlated with time course (cor = 0.14, *p* = 0.01) (Figure 3A,C, Appendix A), but the greenyellow module showed a negative correlation (cor = −0.36, *p* = 0.003) (Figure 3A,D, Appendix A). The salmon (cor = 0.15, *p* = 0.01), purple (cor = 0.19, *p* = 0.001), skyblue (cor = 0.15, *p* = 0.008), and orange (cor = 0.14, *p* = 0.01) modules exhibited a significantly positive correlation with MOI (Figure 3B,E–H, Appendix A).

### 3.2. Biological Functions and Pathways Involved in the Significant Modules

The genes in the salmon module were exactly enriched for influenza A, innate immune response, interferon signaling pathway, and NF-kappa B signaling pathway (Figure 4A,B). Similarly, the genes in the purple module were also enriched for influenza A and NF-kappa B signaling pathways, but involved in inflammatory response (Figure 4C,D).

The skyblue, orange, and greenyellow modules are related to neurotransmitter transport, transmembrane transport, and microbial infection, respectively (Figure 4E–J).

### 3.3. Identification of Critical Genes

To explore hub genes, the PPI network analysis and cytoHubba plugin within Cytoscape were performed based on the rank of the connection degree (number) of each gene. As shown in Figure 5, the top 3 ranked nodes with connection degrees in the PPI network of 5 modules were identified serving as top hub genes. Of the 15 top hub genes, the expression of 13 genes was correlated with the time course and/or MOI in the training group (Figure 6A). In the validation group, the expression of 7 genes has a confirmed correlation with the time course and/or MOI, including BCL2L13 and PLSCR1 in the salmon module, ARID5A and LMO2 in the purple module, NDRG4 in the skyblue module, HAP1 in the orange module, and CARD10 in the greenyellow module (Figure 6B).

## 4. Discussion

In this study, we systematically analyzed a large amount of RNA sequencing data from 8 molecular studies of IAV infection covering test time points between 3 h and 48 h as well as MOI between 0.01 and 5 (see Table 1). We first performed gene co-ex-pression analysis to identify 34 modules. Then, module eigengene change in each experiment design of these independent studies was analyzed using the Limma R package, which was able to avoid bias caused by different experiment conditions. Distribution of the module eigengene change showed that time course and/or MOI guided a correlation with several modules. Further analysis confirmed that five gene modules were significantly correlated with the time course and/or MOI. The genes in two of them were exactly enriched for the influenza A pathway. Thereout, obtaining the above results was absolutely not an accidental event.

The salmon module, one of the two modules, was positively correlated with either time course or MOI. The genes in this module were directly enriched for influenza A, interferon signaling pathway, and innate immune response. Several studies demonstrated that IAV PB1-F2 protein translocated to mitochondria via its interaction with TUFM, and subsequently mediated the engulfment of mitochondria into autophagosome by binding to LC3B, thereby promoting the clearance of MAVS protein, thus negatively modulating the MAVS-mediated antiviral innate immune signaling [21,22].

The LC3B mediates the engulfment of mitochondria into autophagosome through their association with mitophagy effectors, such as the receptors BCL2L13 in mammals [18]. In this study, BCL2L13 was a top hub gene in the salmon module and its expression was positively correlated with the time course and MOI, confirmed in the validation group. In the PPI network of a module, the more neighbors a node has, the more important it is [23,24]. Therefore, BCL2L13 certainly plays a critical role when IAV PB1-F2 protein induces mitophagy to impair innate immunity. On the contrary, another top hub gene, PLSCR1, as a natural antiviral regulator, has been shown to inhibit influenza virus replication in infected cells through interacting with NP of IAV [25,26].

Similarly, the genes in the purple module were also enriched for influenza A and NF-kappa B signaling pathways, but involved in an inflammatory response. Top hub gene ARID5A is able to selectively stabilize STAT3 mRNA to direct naive cells to differentiate into inflammatory cells [27]. Another top hub gene LMO2 transcription start site is located approximately 25 kb downstream from the 11p13 T-cell translocation cluster (11p13 etc.), where a number of T-cell acute lymphoblastic leukemia-specific translocations occur. It was reported that LMO2 took part in an immune response [25]. Several studies indicated that the expression of the two genes was significantly increased in virus-infected cells [28,29,30]. In our study, their expression was only positively correlated with MOI. These suggested that the two genes were highly important in the inflammatory response to IAV.

The genes in the skyblue module were enriched for neurotransmitter transport, including histamine, serotonin, dopamine, etc. It has been shown that host cells can be directly activated in response to IAV, releasing mediators such as histamine, serotonin, and dopamine, which participate in the excessive inflammatory and pathological response observed during IAV infections [31]. N-myc downstream-regulated gene 4 (NDRG4) belongs to the NDRG family, the members of which are expressed in a variety of human organs, and are associated with a wide range of biological processes, such as organ development, tumor inhibition, angiogenesis, and growth regulation [32]. Recently, it was reported that NDRG4 involved in inflammatory mediators release [33]. In the present study, the expression of NDRG4 as a top hub gene in the skyblue module was negatively correlated with MOI. It was believed that NDRG4 was a critical gene in IAV-infected cells releasing mediators.

The virus enters the host cell via transmembrane transport, such as endocytosis and vesicle trafficking. HAP1 gene encoding a protein interacting with cytoskeletal proteins and kinase substrates indicates that the protein plays a role in endocytosis and vesicle trafficking [34]. In the present study, the genes in the orange module were enriched for transmembrane transport. HAP1, as a top hub gene in this module, was positively correlated with MOI. Thus, this gene was bound to play a pivotal role in IAV entry.

The genes in the greenyellow module were enriched for microbial infection. Host response to microbial infection is sensed by sensors such as RIG-I, which induces MAVS-mediated NF-κB and IRF3 activation to promote inflammatory and antiviral responses, respectively [35]. CARMA3, a CARD10 coding scaffold protein, either positively regulates MAVS-induced NF-κB activation or sequesters MAVS from forming high-molecular-weight aggregates, thereby suppressing TBK1/IRF3 activation [36]. When challenged with IAV, CARMA3-deficient mice showed reduced disease symptoms compared to those of wild-type mice due to less inflammation and a stronger ability to clear the infected virus [36]. In our study, the expression of CARD10 as a top hub gene in this module was negatively correlated with time course. This suggested that the expression of CARD10 was initially increased to promote inflammatory responses and was then decreased to unfreeze antiviral responses.

## 5. Conclusions

By a variety of bioinformatics methods (WGCNA, different analysis, PPI, correlation analysis, etc.), this study uncovered and validated fundamental patterns of molecular responses, intrinsic structures of gene co-regulation, and potential targets in influenza A virus infection. Our findings provided further insights into functional investigations to identify potential therapeutic targets against influenza A virus infection.

## Figures and Tables

**Figure 1 viruses-14-01625-f001:**
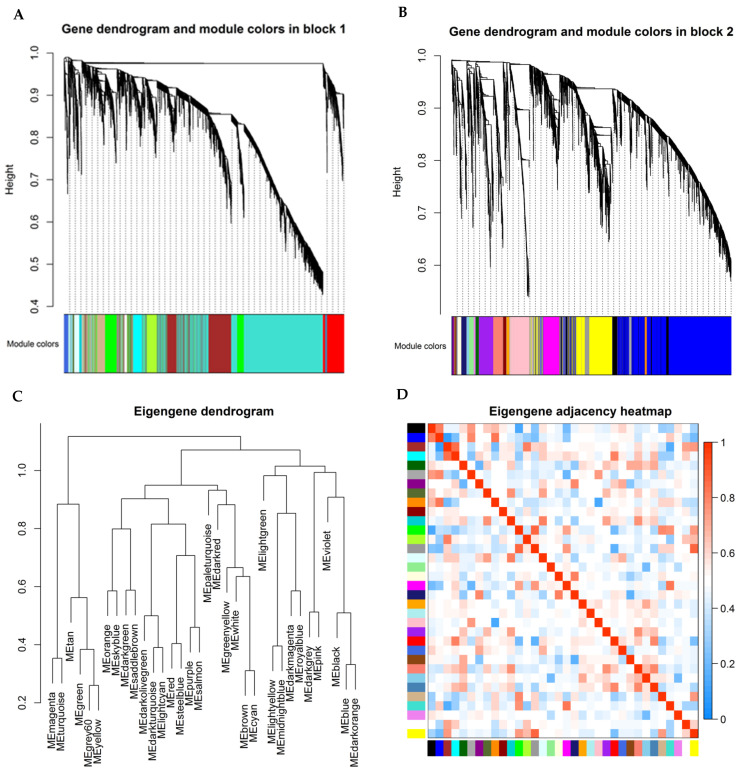
Module detection and network heatmap plot construction using WGCNA in the training group. Each colored row represents a color-coded module that contains a group of highly connected genes. (**A**,**B**) Dendrogram obtained by hierarchical clustering of genes based on their topological overlap is shown at the top; (**C**) Dendrogram of module eigengenes obtained from WGCNA on the correlation; (**D**) Module eigengene adjacency heatmap (each colored row in x- or y-axis represents a color-coded module).

**Figure 2 viruses-14-01625-f002:**
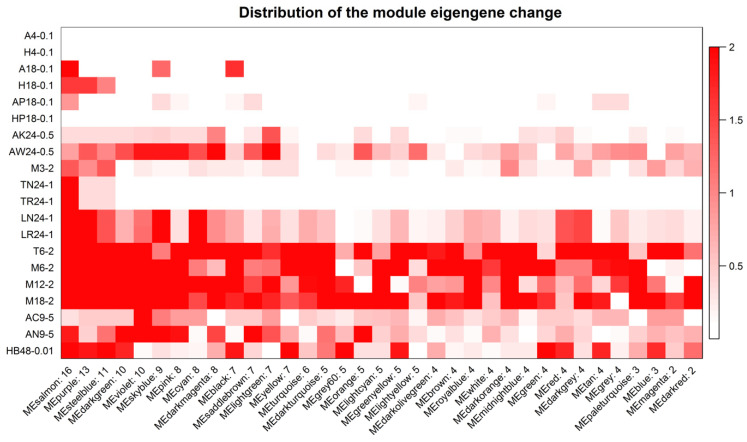
Distribution of module eigengene change in each experiment design of training group. The red graduates with –log (*p*-value), which the maximum is limited to 2. X label: module eigengene and number of *p* < 0.05. Y label: cell type, time point, and MOI (Table 1).

**Figure 3 viruses-14-01625-f003:**
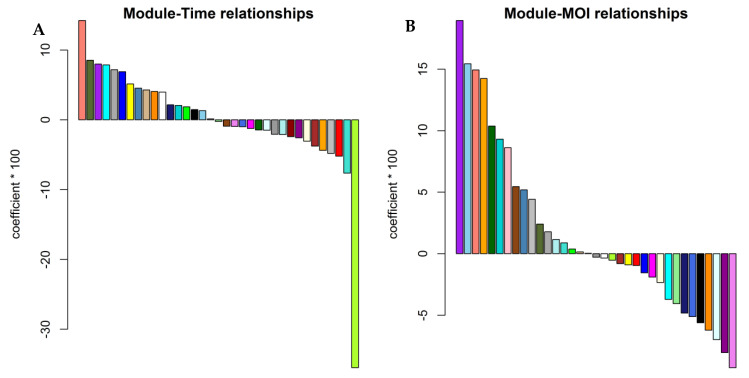
The relationships between module eigengenes and traits. The colors represent modules. (**A**) Correlation of module eigengenes with time course (height: coefficients × 100); (**B**) Correlation of module eigengenes with MOI (height: coefficients × 100); (**C**) Scatter plots of module eigengenes in the salmon module (for Time); (**D**) Scatter plots of module eigengenes in the greenyellow module; (**E**) Scatter plots of module eigengenes in the skyblue module; (**F**) Scatter plots of module eigengenes in the salmon module (for MOI); (**G**) Scatter plots of module eigengenes in the purple module; (**H**) Scatter plots of module eigengenes in the orange module.

**Figure 4 viruses-14-01625-f004:**
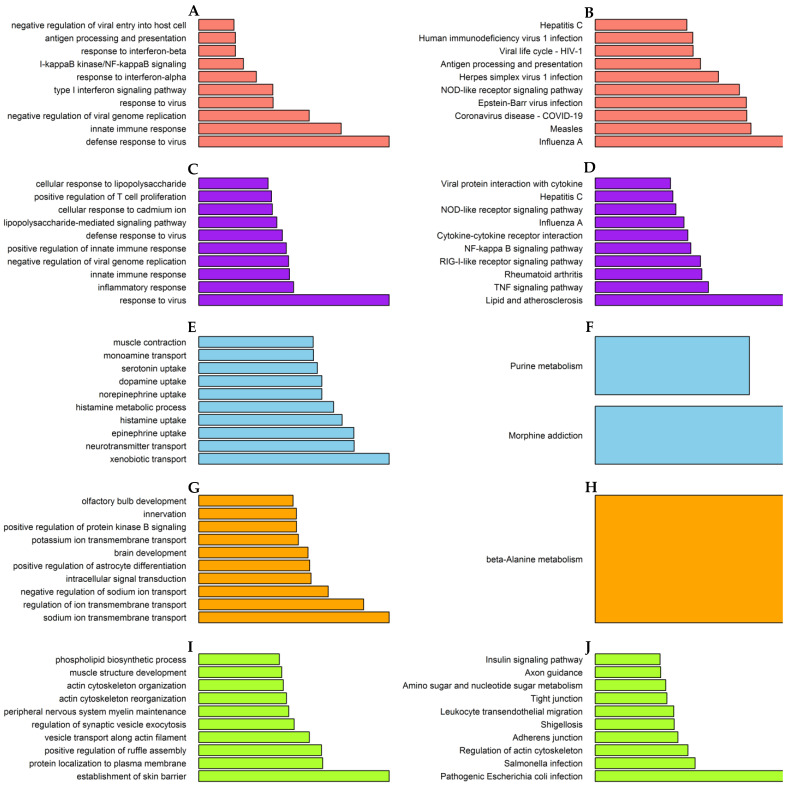
GO and KEGG enrichment analysis of module genes in the training group. The length of red bars displays –log10(*p*-value). (**A**) Salmon module: GO; (**B**) Salmon module: KEGG; (**C**) Purple module: GO; (**D**) Purple module: KEGG; (**E**) Skyblue module: GO; (**F**) Skyblue module: KEGG; (**G**) Orange module: GO; (**H**) Orange module: KEGG; (**I**) Greenyellow module: GO; (**J**) Greenyellow module: KEGG.

**Figure 5 viruses-14-01625-f005:**
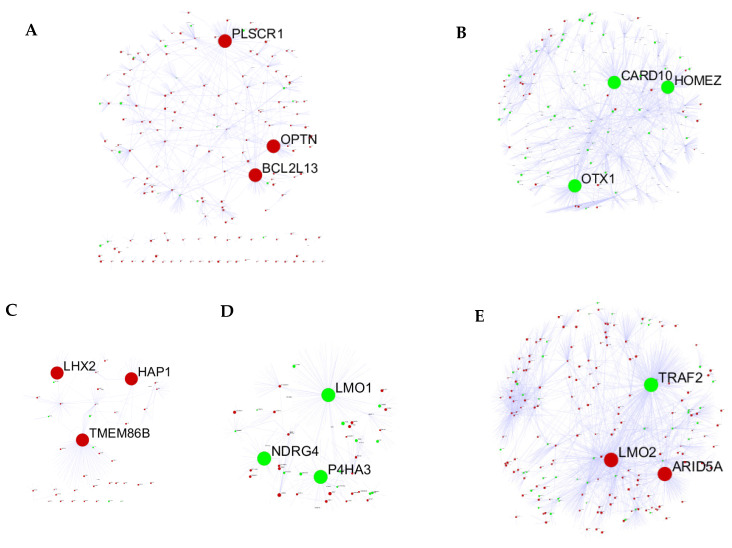
The PPI networks of IAV-related modules. The clearly labeled node was the top hub gene. The red: positive coefficient; the green: negative coefficient. (**A**) Salmon module; (**B**) Greenyellow module; (**C**) Orange module; (**D**) Skyblue module; (**E**) Purple module.

**Figure 6 viruses-14-01625-f006:**
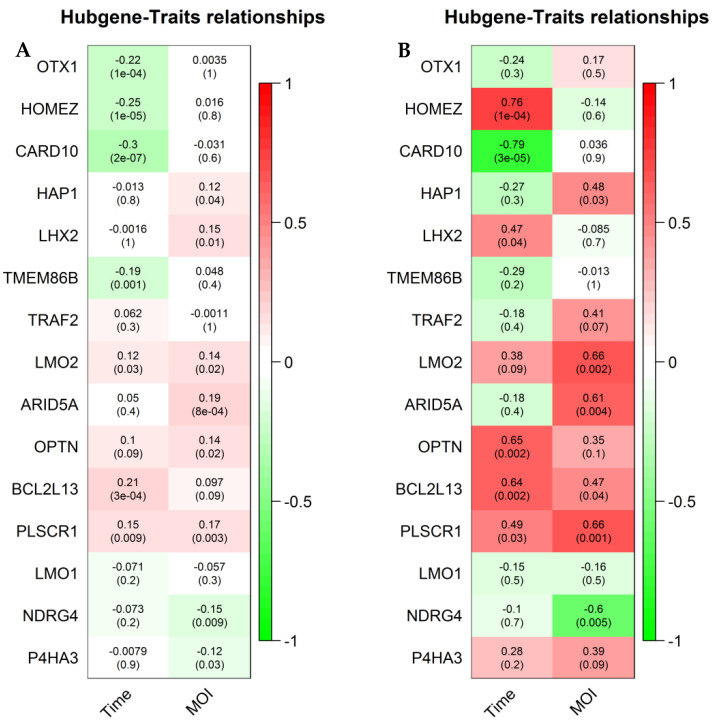
The relationships between top hub genes and traits. The coefficients were the above numbers and the *p*-values were the below numbers. (**A**) In the training group; (**B**) In the validation group.

**Table 1 viruses-14-01625-t001:** The main features of 8 selected datasets in this analysis.

GEO Datasets	Size	Cell Type	Hours	MOI	Abbreviations of Experiment Design
GSE165340	6	A549	9	5	AC9-5; AN9-5
GSE97672	16	MDM	3; 6; 12; 18	2	M3-2; M6-2; M12-2; M18-2
GSE104168	18	A549WT; A549KO	24	0.5	AW24-0.5; AK24-0.5
GSE156152	6	293T	6	1	T6-1
GSE163959	64	Turbinate; Lung	24	1	TN24-1; TR24-1; LN24-1; LR24-1
GSE193164	6	HBEC	48	0.01	HB48-0.01
GSE186908	162	HAE	4; 18	0.1	A4-0.1; H4-0.1; A18-0.1; H18-0.1; AP18-0.1; HP18-0.1
GSE89008	20	HTBE	3; 6; 12; 18; 24	5	HT3-5; HT6-5; HT12-5; HT18-5; HT24-5

## Data Availability

All the data needed to generate the conclusions made in the article are present in the article itself and/or the Appendix A data. Additional data related to this article may be requested from the authors.

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
