# Peer review of "Identification of Critical Genes and Pathways for Influenza A Virus Infections via Bioinformatics Analysis"

_viruses, 2022, doi:10.3390/v14081625_

Round 1

Reviewer 1 Report

Major:

As mentioned by the authors line 55, such in-silico screening aims to identify "novel targets for clinical treatment". However no such thing is performed here by Chen et al..The manuscript is interesting and the application could be significant, however, as of yet, nothing is experimentally validated: no clear hits are defined and experimentally validated.

Validation need to be perform in a cell culture system. Authors need to experimentally validate the correlation with respect to TIME or/and MOI of their "hit"during influenza infection in a cell culture model AND show the therapeutic potential of such target (i.e. use known inhibitor/siRNA/etc to confirm a phenotype).

Minor:

- Overall Figures 1, 2 and 3 are gone through extremely fast. Manuscript would benefit from increased description of the analysis performed.

- Figure 4 should respect color coding as previous figures.

Author Response

Dear reviewer,

Thank you so much for your comments on our manuscript entitled “Identification of Critical Genes and Pathways for Influenza A Virus Infections via Bioinformatics Analysis" (ID: viruses-1811208). Those comments are very helpful for revising and improving our paper, as well as the important guiding significance to other research. We have studied the comments carefully and made corrections which we hope meet with approval. The main corrections are in the manuscript and the responses to the reviewers' comments are as follows (the replies are highlighted in red).

  • As mentioned by the authors line 55, such in-silico screening aims to identify "novel targets for clinical treatment". However, no such thing is performed here by Chen et al..The manuscript is interesting and the application could be significant, however, as of yet, nothing is experimentally validated: no clear hits are defined and experimentally validated.

Validation need to be perform in a cell culture system. Authors need to experimentally validate the correlation with respect to TIME or/and MOI of their "hit"during influenza infection in a cell culture model AND show the therapeutic potential of such target (i.e. use known inhibitor/siRNA/etc to confirm a phenotype).

Response 1:

We want to thank you for your constructive and insightful criticism and advice. We agree that more studies would be useful to further validate the correlation with respect to TIME or/and MOI. Indeed, it will be more profound if we get the relevant results. However, these studies are beyond the scope of this report which focuses on answering critical questions regarding the identification of the critical genes and pathways for Influenza A Virus infections via bioinformatics analysis. We note also that since the submission of our manuscript there has been a publication by Forst CV showing that DOCK5’s targets determined by the DOCK5 knockout experiments strongly validated the gene signatures and networks (Ref. 10). We have added this citation and reference to the manuscript.

As the entitle of this manuscript, we aim to identify the critical genes and pathways for Influenza A Virus infection. In this study, we firstly performed WGCNA to investigate modules correlated with TIME or/and MOI. Then, combined with the PPI network analysis, top hub genes were identified in these modules. According to the Centrality-Lethality Rule (In the PPI network of a module, the more neighbors a node has, the more important it is), certainly, the top hub genes correlated with TIME or/and MOI were the critical genes for Influenza A Virus infections. These correlations were validated in another independent experiment (GSE89008). Integrating these analyses was able to identify critical genes.

  • Overall Figures 1, 2 and 3 are gone through extremely fast. Manuscript would benefit from increased description of the analysis performed.

Response 2:

We gratefully appreciate your valuable suggestion. We have increased the description of the analysis performed. Please see page 3 of the revised manuscript, lines 119–130, and page 4, lines 138–145.

  • Figure 4 should respect color coding as in previous figures.

Response 3:

Thanks for your nice suggestion. According to the revised content, we have redrawn Figure 4 to clearly show those modules and replaced the previous Figure 4 in the revised manuscript. Please see page 7 of the revised manuscript.

Once again, thank you very much for your constructive comments and suggestions which would help us in depth to improve the quality of the paper.

Kind regards.

Sincerely yours,

Gao Chen (First author)

Email: gaochen_npwr@hotmail.com

Jing Zhao (Corresponding author)

Email: hhzhaojing@hotmail.com

Reviewer 2 Report

This work is well-designed and deserves the attention of the scientific community.

There are some factors that would improve the clarity of the paper, listed below.

  • Figure S1: please label the x and y-axises. It’s difficult to know whether the authors removed the batch effect from this figure without labels and legend.

  • Line #86: cite the WGCNA package

  • Line #97-99: cite the online tool David, Gene Ontology database, and KEGG

  • Line #119: Can you explain what’s the difference between Figure 1A and 1B?

  • Figure 1C: lack of y-axis label; Figure 1D: lack of label of color legend

  • Line 122-124: it’s difficult to have the statement of correlation between modules and IAV strains/cell type from Figure 2. It’s not straightforward to get the information about strains and cell types from abbreviations. Can you please use different colors or texts to label the y-axis? 

  • Figure 3A and 3B: lack of y-axis label; Figure 3C and 3F: same figure legend. Please provide more information to distinguish between these two.

  • Figure 4: plot the x-axis ticks and add vertical lines showing p-value cutoff: x-intercept = -log10(0.05).

  • Figure 5: lack of color labels. What does the color red and green represent, respectively?

  • Line #220: Figure 6 shows the contradictory results as the authors stated here. The coefficients of NDRG4 are negative, but authors stated that NDRG4 is positively correlated with MOI. It's suspicious.

  • Figure 6: some genes showed the opposite results from training and validation groups, for example, LHX2. The authors should discuss why this happens and how confident they stated that these top hub genes are correlated with time course and MOI.

Author Response

Dear reviewer,

Thank you for your comments on our manuscript entitled “Identification of Critical Genes and Pathways for Influenza A Virus Infections via Bioinformatics Analysis" (ID: viruses-1811208). Those comments are very helpful for revising and improving our paper, as well as the important guiding significance to other research. We have studied the comments carefully and made corrections which we hope meet with approval. The main corrections are in the manuscript and the responses to the reviewers' comments are as follows (the replies are highlighted in red).

This work is well-designed and deserves the attention of the scientific community.

We thank you for reading our manuscript carefully and giving the above positive comments.

  • Figure S1: please label the x and y-axises. It’s difficult to know whether the authors removed the batch effect from this figure without labels and legend.

Response 1:

Thank you for the above suggestion. We think this is an excellent suggestion. We have added the legend of figure S1 in the revised supplementary materials, describing labels of the x and y-axises. We hope that it is now clearer. Please see page 1 of the revised supplementary materials.

  • Line #86: cite the WGCNA package

Response 2:

We thank you for pointing out this issue. We have added the reference [14] to subsection 2.2 in the revised manuscript. Please see page 3 of the revised manuscript, line 85.

  • Line #97-99: cite the online tool David, Gene Ontology database, and KEGG

Response 3:

Thanks for your nice suggestion. Following your suggestion, we have added the reference [16-18] to subsection 2.3 in the revised manuscript. Please see page 3 of the revised manuscript, lines 97-98.

  • Line #119: Can you explain what’s the difference between Figure 1A and 1B?

Response 4:

We thank you for pointing out this issue.

“In order to resolve computational challenges caused by constructing and analyzing networks with such large numbers of nodes, function blockwiseModules in the WGCNA package were used to split into two blocks for hierarchical clustering. The hierarchical clustering trees were constructed following a dynamic hybrid cut (Figure 1A, B)”. We have inserted this description into subsection 3.1 in the revised manuscript. Please see page 3 of the revised manuscript, lines 119–123.

  • Figure 1C: lack of y-axis label; Figure 1D: lack of label of color legend

Response 5:

We are very sorry for our negligence with the color legend. Thank you for pointing out the problems. The y-axis in Figure 1C is height by default and is not labeled in the WGCNA package. According to the revised content, we have inserted the description of color into the legend. Please see page 4 of the revised manuscript, line 135.

  • Line 122-124: it’s difficult to have the statement of correlation between modules and IAV strains/cell type from Figure 2. It’s not straightforward to get the information about strains and cell types from abbreviations. Can you please use different colors or texts to label the y-axis? 

Response 6:

We sincerely thank you for your careful reading and helpful comments. Thank you for pointing out the above problems. We have inserted the statement about IAV strains/cell types into subsection 3.1 in the revised manuscript. Please see page 3 of the revised manuscript, line 130.

“Among the 7 studies in the training group, all of IAV strains and most of the cell types were different (Table 1)”. Please see page 4 of the revised manuscript, lines 138-139.

We have carefully considered all comments and revised our manuscript accordingly. However, the color-coded experiment is easy to confuse the color-coded module, and the text labeling the y-axis is too long. So, we think that we may only use abbreviations to label the y-axis.

  • Figure 3A and 3B: lack of y-axis label; Figure 3C and 3F: same figure legend. Please provide more information to distinguish between these two.

Response 7:

We are very sorry for our negligence with labels. Thank you for pointing out the two problems. According to the revised content, we have redrawn Figures 3A and 3B to clearly show the y-axis label and added the description to distinguish between the two in the legend. There are gene significance for Time in Figure 3C and gene significance for MOI in Figure 3F. Please see page 6 of the revised manuscript, lines 159 and 161.

  • Figure 4: plot the x-axis ticks and add vertical lines showing p-value cutoff: x-intercept = -log10(0.05).

Response 8:

Thanks for your nice suggestion. We selected items ranked within the top 10, of which all of the p-values were less than 0.05. We have inserted this description into subsection 2.3 in the revised manuscript. Please see page 3 of the revised manuscript, lines 100. 

  • Figure 5: lack of color labels. What does the color red and green represent, respectively?

Response 9:

We are very sorry for our negligence with color labels. Thank you for pointing out this issue. We have inserted the description” The red: positive coefficient; the green: negative coefficient” into the legend in the revised manuscript. Please see page 8 of the revised manuscript, line 210.

  • Line #220: Figure 6 shows the contradictory results as the authors stated here. The coefficients of NDRG4 are negative, but authors stated that NDRG4 is positively correlated with MOI. It's suspicious.

Response 10:

We were really sorry for our careless mistake. Thank you for pointing this out. NDRG4 is negatively correlated with MOI. We have corrected the errors in the revised manuscript. Please see page 9 of the revised manuscript, line 233.

  • Figure 6: some genes showed the opposite results from training and validation groups, for example, LHX2. The authors should discuss why this happens and how confident they stated that these top hub genes are correlated with a time course and MOI.

Response 11:

Thanks for your rigorous consideration. We think that the probability caused the opposite phenomenon under the different experimental conditions. So, we believe that consistent result from training and validation groups is confident, although we cannot offer assurance one hundred percent.

Once again, thank you very much for your constructive comments and suggestions which would help us in depth to improve the quality of the paper.

Kind regards.

Sincerely yours,

Gao Chen (First author)

Email: gaochen_npwr@hotmail.com

Jing Zhao (Corresponding author)

Email: hhzhaojing@hotmail.com

Round 2

Reviewer 1 Report

Authors made sufficient efforts to address my comments.

Reviewer 2 Report

The manuscript was much improved according to the revision and the authors addressed the required points adequately.